# Protective Application of *Morus* and Its Extracts in Animal Production

**DOI:** 10.3390/ani12243541

**Published:** 2022-12-15

**Authors:** Lixue Wang, Huaqi Gao, Cui Sun, Lingxia Huang

**Affiliations:** 1College of Animal Sciences, Zhejiang University, Hangzhou 310058, China; 2Institute of Fruit Science, College of Agriculture and Biotechnology, Zhejiang University, Hangzhou 310058, China; 3Key Laboratory of Silkworm and Bee Resource Utilization and Innovation of Zhejiang Province, Zhejiang University, Hangzhou 310058, China

**Keywords:** mulberry, extracts, animal production, active compounds, feed additives

## Abstract

**Simple Summary:**

Mulberry tree leaves are regarded as a potential protein resource for livestock feeding. They also contain many active compounds that confer a variety of bioactive functions, such as antioxidant and anti-inflammatory properties. Although the beneficial effects of mulberry tree products as natural feed additives have been reported in ruminants, their application in other farm animal production needs to be further investigated. Here we summarized the research progress on the use of the mulberry and its extracts on the growth performance and health of animals (poultry, swine, ruminants, and fishes). The biological functions of the mulberry tree and its extracts are briefly described, which could provide a reference for future research and applications of the mulberry tree in animal production.

**Abstract:**

Different components of the mulberry tree (fruits, leaves, twigs, and roots) are rich in active compounds, and have been reported to possess potent beneficial properties, including antioxidative, anti-inflammatory, antimicrobial, anticancer, anti-allergenic, antihypertensive, and neuroprotective. The mulberry and its extracts can effectively improve the growth performance and fitness of animals. They not only possess the properties of being safe and purely natural, but also they are not prone to drug resistance. According to the literature, the supplemental level of the mulberry and its extracts in animal diets varies with different species, physiological status, age, and the purpose of the addition. It has been observed that the mulberry and its extracts enhanced the growth performance, the quality of animal products (meat, egg, and milk), the antioxidant and the anti-inflammatory responses of animals. Furthermore, the mulberry and its extracts have antibacterial properties and can effectively moderate the relative abundance of the microbial populations in the rumen and intestines, thus improving the immunity function of animals and reducing the enteric methane (CH_4_) production in ruminants. Furthermore, the mulberry and its extracts have the potential to depurate tissues of heavy metals. Collectively, this review summarizes the nutrients, active compounds, and biological functions of mulberry tree products, as well as the application in livestock production with an aim to provide a reference for the utilization of the mulberry and its extracts in animal production.

## 1. Introduction

Population growth, urbanization, and rising incomes have led to the dramatic demand for animal products [1]. The global demand for animal products will rise by more than two-thirds by the year 2050 [2]. With the increasing demand for animal products, we have to explore new non-conventional feed resources to ensure the sustainability of animal production [3,4]. A variety of alternative feed resources exist for livestock, such as crop residues, the leaves of shrubs and trees, and weeds. However, these alternative feed resources have a lower digestibility, a lower content of protein and energy, and higher antinutritional factors, which limit their application [5,6].

The mulberry tree belongs to the *Morus* genus of the Moraceae family, and is distributed all over the world [7]. The mulberry tree contains 24 species and one subspecies. Among them, *M. albus*, *M. atropurpurea*, *M. multicaulis*, and *M. bombycis* are the dominant species [8]. The mulberry tree originated in China, Japan, and the Himalayan foothills. China possesses the most mulberry land with over 626,000 ha, followed by India with around 280,000 ha [9].

The mulberry tree is rich in bioactive compounds, including polysaccharides, phenols, flavonoids, and alkaloids, and has been reported to possess potent beneficial properties, including antioxidative, antidiabetic, and anti-cholesterol [10]. All parts of the mulberry tree, including the leaves, fruits, stems, and roots, are used for various purposes [11]. In addition, mulberry leaves are an excellent source of protein for livestock, with 14.0–34.2% protein content [12,13]. Extensive studies have demonstrated that mulberry leaves are a high quality protein source in the diets of animals, including pigs [14], hens [15], sheep [16], and cattle [17]. However, the effects of the mulberry tree and its extracts on animals are dependent on several factors, such as animal species, level of supplementation, the method for using the mulberry, and farm management. Individual studies cannot take into account all of these variables. This review summarizes the nutrients, active compounds, and biological functions of mulberry tree extracts, on the growth performance and fitness of animals. Our objective is to assess the beneficial effects of the mulberry tree and its extracts along with their importance in animal production, and provide a reference for future research and applications. 

## 2. Nutrients Derived from the Mulberry Tree and Its Extracts

The mulberry tree, a member of the family Moraceae and genus *Morus* [18], is widely distributed throughout the world [19]. *Morus alba* (white mulberry), *Morus nigra* (black mulberry), and *Morus rubra* (red mulberry) are the most common species [20]. The mulberry tree is a potential protein source for animals. Different parts of the mulberry, especially the leaves and fruits, contain a variety of nutrients [21]. 

### 2.1. Leaves

The nutritional composition of the mulberry leaves is influenced by many factors, such as the varieties, environments, ecologies, and harvest conditions, and the nutritional composition varies greatly in different studies. All of the reported nutrient components in the mulberry leaves are displayed in Table 1. Fresh mulberry leaves contain dry matter (DM, 19.8–30.40%), a substantial amount of crude protein (CP, 4.72–22.3%), fats (0.64–4.36%), crude fiber (CF, 5.26–15.9%), total ash (4.10–14.50%), carbohydrates (carb, 8.01–13.42%), neutral detergent fiber (NDF, 8.15–43.4%), and gross energy (GE, 69–224 kcal/100 g) [22,23,24,25,26]. Moreover, according to Srivastava et al. [22], mulberry leaves are a plentiful source of important minerals and vitamins, such as calcium (Ca, 380–786 mg/100 g), ascorbic acid (200–280 mg/100 g), β-carotene (10,000–14,688 μg/100 g), iron (Fe, 4.7–10.36 mg/100 g), zinc (Zn, 0.22–1.12 mg/100 g), and tannic acid (0.04–0.08%). 

As for dried mulberry leaves powder, it contains DM (18.0–95.5%), CP (11.75–37.36%), fats (2–11.10%), CF (5.4–32.3%), nitrogen free extract (NFE, 42.2–54%), NDF (19.38–36.66%), acid detergent fiber (ADF, 10.2–29.7%), total ash (7.56–22.36%), carb (9.7–56.42%), and GE (113–422 kcal/100 g) [22,25,27,28,29,30,31,32,33,34,35,36,37,38]. Dried mulberry leaves powder also possesses Ca (137.5–2226 mg/100 g), ascorbic acid (100–200 mg/100 g), β-carotene (8438–13,125 μg/100 g), Fe (14.15–35.72 mg/100 g), Zn (0.72–5.75 mg/100 g), and tannic acid (0.12–0.76%) [22,27,31,32,36].

**Table 1 animals-12-03541-t001:** Chemical composition of mulberry leaves.

Authors	Species	Source	Country	Season	DM, %	CP, %	Fat ^1^, %	CF, %	NFE, %	NDF, %	ADF, %	Ash, %	Carb, %	GE, kcal/100 g
Yao et al., 2000	*Morus alba* L.	Fresh	China	Spring and Autumn	23.6–30.4	19.6–21.9	-	-	-	37.5–43.4	-	-	-	-
Srivastava et al., 2006	*Morus alba* L.	Fresh	India	Spring	23.32–28.87	4.72–9.96	0.64–1.35	-	-	8.15–11.32	-	4.26–5.32	8.01–13.42	69–79
		Dried leaf powder			92.76–94.89	15.31–30.91	2.09–4.93	-	-	27.6–36.66	-	14.59–17.24	9.7–29.64	113–224
Todaro et al., 2009	*Morus latifolia*	Fresh	Italy	Summer	25.98	21.05	4.36	-	-	22.88	19.72	13.31	-	-
Adeduntan et al., 2009	*Morus alba* L.	Dried leaf powder	Nigeria	-	20.65–27.84	21.24–21.66	5.31–8.02	8.74–13.7	-	-	-	8.19–12.63	47.27–56.42	-
Kandylis et al., 2009	*Morus alba* L.	Dried leaf powder	Greece	-	89.4	15.1	2	18.9	54	-	-	10	-	-
Al-Kirshi et al., 2009	*Morus alba* L.	Dried leaf powder	Malaysia	-	89.3	29.8	11.1	32.3	-	22.8	22.8	11.8	-	422
Vu et al., 2011	*Morus alba* L.	Fresh	The Netherlands	-	19.8	22.3	3.5	15.9	-	31.1	18.3	14.5	-	-
Sahoo et al., 2011	*Morus alba* L.	Dried leaf powder	India	-	27.8	19.4	4.1	-	-	36.1	26.8	13.3	-	-
Guven, 2012	*Morus nigra*	Dried leaf powder	Turkey	Summer	42.2	16.06	-	-	-	22.08	19.46	17.5	-	-
	*Morus alba*				46.27	18.73	-	-	-	19.38	17.33	15.4	-	-
	*Morus rubra*				37.36	11.75	-	-	-	33.33	24.06	22.36	-	-
	*Morus alba pendula*				25.97	23.72	-	-	-	29.53	26.06	17.7	-	-
Wang et al., 2012	*Morus atropurpurea Roxb*	Dried leaf powder	China	Summer	-	25.17	2.85	-	-	27.88	16.49	-	-	-
	*Morus alba* L.				-	25.9	4.21	-	-	26.25	17.07	-	-	-
	*Morus multicaulis Perr*				-	25.18	4.91	-	-	27.54	17.66	-	-	-
Iqbal et al., 2012	*Morus alba* L.	Dried leaf powder	Pakistan	-	94.7	18.41	6.57	10.11	-	-	-	8.91	-	-
	*Morus nigra* L.				93.3	19.76	5.13	12.32	-	-	-	9.12	-	-
	*Morus rubra* L.				95.5	24.63	4.24	8.17	-	-	-	11.73	-	-
Flaczyk et al., 2013	*Morus alba* L.	Aqueous extracts	Poland	-	94.6	12.7	0.15	-	-	-	-	22.7	-	-
Dolis et al., 2017	*M. multicaulis*	Fresh	Romania	Summer	-	6.2	1.04	5.26	12.77	-	-	4.1	-	-
		Dried leaf powder			-	21.16	3.54	17.88	43.46	-	-	13.96	-	-
Yu et al., 2018	*Morus alba* L.	Dried leaf powder	China	Summer	-	29.02–37.36	-	13.01–16.61	-	-	-	-	-	-
	*M. multicaulis Perr.*				-	27.63–36.42	-	11.46–15.27	-	-	-	-	-	-
	*M.atropurpurea Roxb.*				-	28.29–34.19	-	12.41–15.50	-	-	-	-	-	-
Cai et al., 2019	*Morus alba* L.	Dried leaf powder	China	-	18–27	17–19.4	-	-	-	21.8–27.8	10.2–13	10.8–12.8	-	-
Kang et al., 2020	-	Dried leaf powder	China	Spring and Autumn	-	24.8	5.4	-	42.2	33.6	25.8	-	-	-
					-	20.9	7	-	47.2	26.6	18.1	-	-	-
					-	26.9	6.3	-	43.1	25.2	29.7	-	-	-
					-	22.4	7.9	-	44.3	31.8	29.7	-	-	-
Ouyang et al., 2019	*Morus alba var. multicaulis*	Dried leaf powder	China	Spring	89.54	20.3	8.15	-	-	34.3	16.28	7.56	-	-

Dry matter (DM); crude protein (CP); crude fiber (CF); nitrogen free extract (NFE), neutral detergent fiber (NDF); acid detergent fiber (ADF); carbohydrate (Carb); gross energy (GE). ^1^ Fat was divided into crude fat and ether extract, both of which were determined by the AOAC methods. The studies of Srivastava et al., 2006 [22], Adeduntan et al., 2009 [27], Iqbal et al., 2012 [33], Flaczyk et al., 2013 [34] and Kang et al., 2020 [37] were described as crude fat. In the rest of studies, they were described as ether extract.

### 2.2. Fruits

The mulberry fruit also contains CP, fats, minerals, and other components, and is a healthy food choice for consumers [39]. Similar to mulberry leaves, the nutritional and chemical composition of the mulberry fruit changes with the varieties, environments, climatic conditions, and soil conditions. The average contents of the trace element components in the mulberry fruits are shown in Table 2. DM can range from 9.45% to 28.50%, CP can vary from 0.51% to 12.98%, fat can vary from 0.34% to 7.21%, CF can vary from 0.57% to 14.0%, ash can vary from 0.46% to 4.79%, and carb can vary from 13.83% to 71.7% [20,40,41,42,43,44,45,46]. A study by Imran et al. [41] showed that the GE in the mulberry fruit can range from 67.36 to 84.22 kcal/100 g.

Additionally, the mulberry fruit also contains Fe (1.17–77.6 mg/100 g), Zn (0.14–59.20 mg/100 g), Ca (38.89–576 mg/100 g), magnesium (Mg, 12.21–360 mg/100 g), kalium (K, 87.70–2170 mg/100 g), sodium (Na, 5.9–280 mg/100 g), manganese (Mn, 0.03–4.36 mg/100 g), and copper (Cu, 0.04–1.31%) [20,40,41,42,43,46,47,48,49]. Ascorbic acid ranges from 15.20 to 22.4 mg/100 g [20,41]

Collectively, the chemical composition of mulberry leaves and mulberry fruits varies greatly due to the different varieties, geographical environment, seasons, and other factors. The results suggest that we should pay attention to the regional and variety differences of the mulberry tree’s raw materials, when using mulberry tree products in feed production.

## 3. Bioactive Compounds in the Mulberry and Their Bioactivities

The history of the mulberry tree and its extracts used as a medicinal herb is very long, due to its extensive biological and pharmacological activities [50]. Mulberry plants contain a variety of compounds with medical and veterinary pharmacological properties, including alkaloids, flavonoids, phenolic acids, and others. These compounds have antioxidant, anti-inflammatory, antibacterial, anticancer, antidiabetic, neuroprotective, cardioprotective, hepatoprotective, antihypertensive, anti-apoptosis, antiviral, anti-arteriosclerosis, and antidepressant properties (Figure 1). 

The mulberry tree is identified as an appreciable source of flavonoids, which have beneficial effects on human and animal health. Previous studies reported that the concentrations of the total flavonoids were 9.84–58.42 mg/g in dried mulberry leaves from different varieties [33,35,51,52]. Flavonoids have been reported to exert diverse biological effects, such as anticancer, neuroprotective, hepatoprotective, nephroprotective, antidiabetic, cardio-protection, and antibacterial, mainly associated with their antioxidant and anti-inflammatory activities [53,54]. Further studies showed that the inflammatory and antioxidant effects of flavonoids were mediated through regulating the NF-κB, AP-1, PPAR, Nrf2, MAPKs, JNK, p38, ERK, PI3-K/Akt, and PKC signaling pathways [55,56].

1-Deoxynojirimycin (1-DNJ), a polyhydroxylated piperidine alkaloid [35], is a potent α-glucosidase inhibitor with unique bioactivities, such as anti-inflammatory, antioxidant, and anticancer [57,58,59]. 1-DNJ exhibited anti-hyperglycemic activity through regulating the expression of proteins related to glucose transport systems, glycolysis, and gluconeogenesis enzymes [60,61,62]. In addition, 1-DNJ also possesses anti-microbial properties. Hu et al. [61] reported that the 1-DNJ treatment could promote the growth of beneficial bacteria and suppress the growth of harmful bacteria in a streptozotocin-induced diabetic mouse model. 

Phenolic acids were found to be excellent antioxidant agents. By scavenging free radicals, modulating the antioxidant enzyme activity, and regulating the signaling pathways associated with oxidative stress, the phenolic acids can exert an antioxidant activity [63,64]. Phenolic acids are also known for their anti-inflammatory properties. Oxidative stress and the resulting oxidative damage play an important role in the formation and progression of cancer [65]. To some extent, phenolic acids could inhibit the proliferation of colon cancer cells and induce apoptosis in cancer cells through oxidant-mediated mechanisms [66]. Previous studies demonstrated that phenolic acids could rupture the cell membrane integrity and inhibit the growth of pathogenic bacteria [67,68,69]. The significant antidiabetic effect of phenolic acids may be due to the reduced levels of oxidative stress and pro-inflammatory cytokines [70,71]. Peng et al. [72] found that phenolic acids maintain the glucose homeostasis by regulating the expression of the intestinal glucose transporters and proglucagon.

## 4. Application of Mulberry Tree Extracts in Animal Production

Mulberry leaves are considered a potential feed additive, due to their high protein content [36]. Different parts of the mulberry tree, including the fruits, leaves, twigs, and roots, contain a variety of phytochemicals with a variety of biological activities [73]. Phytochemicals act as prebiotics to improve human (and animal) health by altering the composition of the gut microbes [74]. In addition, mulberry tree extracts are an important Chinese herbal medicine, with significant effects on improving the body immunity, protecting multiple organs (e.g., liver, cardiac, liver, and kidney), as well as fighting oxidative stress and inflammation [75,76,77]. Several studies have documented that the beneficial effects of the mulberry tree extracts’ inclusion in animals’ diets. 

### 4.1. Poultry

Table 3 summarizes the application of the mulberry tree and its extracts in poultry production between 2008 and 2022. Islam et al. [78] evaluated the effect of mulberry leaf powder or its extract on the growth performance and blood cholesterol levels of broilers, and found that the addition of 3.5% mulberry leaf powder or its extract to the diet can improve the broiler body weight and feed conversion efficiency, when compared with the control diet. The supplementation of the mulberry leaf powder or its extract significantly decreased the total cholesterol and triglycerides at 15 to 22 d and 22 to 42 d, compared to the control and antibiotic groups. The mulberry leaf powder or its extract can be used in broiler feed, in order to produce healthy broilers with low cholesterol. Chen et al. [79] examined the effects of adding 4% mulberry leaf powder on the gut microbiota of broilers. The relative abundances of *Bacteroides*, *Prevotella*, and *Megamonas* in the mulberry leaf powder diet were greater than those in the control diet. The supplementation of the mulberry leaf powder in the diet could affect the gut microbiota by changing their compositions. Chen et al. [80] studied the immune-enhancing effects of the mulberry leaf polysaccharides on broilers immunized with the Newcastle disease (ND) vaccine. It was shown that the mulberry leaf polysaccharide improved the serum antibody titer and the concentrations of interleukin-2 (IL-2), interferon-gamma (IFN-γ), and secretory immunoglobulin A (sIgA) in tracheal and jejunal wash fluids, and the quantity of immunoglobulin A-positive cells in the cecal tonsil. In addition, mulberry leaf polysaccharides significantly increased the body weight of broilers.

Mature mulberry leaves are usually high in crude fiber and anti-nutritional factors, which can be decreased by microbial fermentation [22,81]. Ding et al. [82] examined the effects of adding different doses of fermented mulberry leaf powder in the basal diet on the growth performance and meat quality of broilers. Among the different treatments, broilers fed a 3% fermented mulberry leaf powder diet had the highest average daily gain (ADG) and feed conversion efficiency. Compared with the control group, the 3% fermented mulberry leaf powder group has a positive effect on the nutrient digestibility, slaughter performance, and meat quality.

Medicinal herbs are typically used in the form of compound preparations, in proportion [83]. Jang et al. [84] evaluated the effects of a medicinal herb extract mix (MHEM, the mixture of mulberry leaf, Japanese honeysuckle, and goldthread) containing mulberry leaf, on the anti-oxidative potential and quality of broilers’ breast meat. The breast meat composition was not significantly affected by the MHEM-supplemented diet, but the content of the total phenols and the activity of the 1,1-diphenyl-2-picrylhydrazyl (DPPH) radical-scavenging during storage were significantly increased by it. The best overall preference for breast meat was observed in the 0.3% MHEM-supplemented diet throughout the 7-d storage period.

The use of mulberry leaves in the feed not only improved the growth performance of the laying hens, but also improved their antioxidative status and egg quality [85]. Huang et al. [86] studied the beneficial effects of the mulberry leaf flavonoids on the reproduction performance of aged breeder hens and their mechanism. The mulberry leaf flavonoids treatment decreased the level of the total cholesterol (T-CHO), low-density lipoprotein cholesterol (LDL-C), and alkaline phosphatase (AKP) in serum, and increased the high-density lipoprotein cholesterol (HDL-C) concentrations in serum, thus reducing the blood lipid. Dietary mulberry leaf flavonoids improved the antioxidant capacity by increasing the activities of superoxide dismutase (SOD) and decreasing the malondialdehyde (MDA) level in the ovary of aged hens. Mulberry leaf flavonoids significantly increased the serum estradiol levels and the mRNA expression of *CYP19A1*, *ERβ*, and *LHR* in the ovary, and had a positive effect on the ovarian function. In addition, mulberry leaf flavonoids enhanced the fatty acid oxidation by un-regulating the expression of *PPARα* and *CPT-I* in the liver. Huang et al. [87] also showed that the dietary mulberry leaf flavonoids could effectively improve the antioxidative activity and Ca deposition in the uterus shell gland, providing a new method to improve the quality of aged hens’ eggshells. Molecular studies showed that the mulberry leaf flavonoid treatment activated the Nrf2 signaling pathway, thus having a protective effect on the shell gland damage caused by aging.

### 4.2. Swine

Several studies investigated the effect of mulberry tree extracts on swine, as shown in Table 4. Li et al. [88] found that by adding 10% mulberry leaves to the diet, improved the carcass yield and decreased the leaf lard percentage and backfat thickness, while not affecting the growth rate of finishing pigs. The results showed that the 10% mulberry leaf additive significantly increased the content of inosine and fat in the muscle. Further studies revealed that mulberry leaves may regulate the metabolism and deposition of fat, by controlling the activities of sucrase, lipase, and glucose metabolism enzymes in the liver, thus improving the meat quality. Song et al. [89] also reported that the mulberry leaf powder treatment could effectively improve the meat quality and increase the antioxidant activity of the muscles of finishing pigs. Their results revealed that adding 10% mulberry leaf powder could decrease the backfat thickness and cholesterol content in muscles, increase the SOD activity in muscles, and regulate the fat metabolism by reducing the serum triglyceride levels. Liu et al. [14] investigated the effects of adding mulberry leaves at different levels on the growth performance and carcass traits of finishing pigs. Among the different treatments, the mulberry leaves in the diet at <12% could improve the quality of meat and the chemical composition of muscles, but has no negative impact on the growth performance. Mulberry leaves in the diet at <12% also enhanced the serum antioxidant property, increased the polyunsaturated fatty acid content in muscles, and inhibited the lipid oxidation [90]. In addition, Chen et al. [91] revealed that the dietary mulberry leaves improved the quality traits of the muscles in finishing pigs, and explored its molecular mechanisms by transcriptome profiling. The dietary supplementation of 6% dietary mulberry leaves may improve the muscle quality by regulating the expression of several key genes, such as *TNNC1*, *MYL3*, and *TNNT1*. Chen et al. [92] showed that supplementing finishing pigs with 4% mulberry leaf powder not only enhanced the growth performance, but also improved the carcass traits and the quality of the pork. Moreover, mulberry leaf powder improved the meat flavor by increasing the content of CP, amino acids, and the total unsaturated fatty acids (TUFAs) as well as decreasing the concentrations of the total saturated fatty acids (TSFAs). However, Zeng et al. [93] found that the supplementation of mulberry leaf powder (15%) improved the quality of meat from finishing pigs, but decreased the growth performance and carcass traits. It may improve the quality of the meat by changing the myofiber characteristics, enhancing the antioxidant capacity, and increasing the intramuscular fat.

Mulberry leaf extracts can enhance the growth performance and the antioxidant capacity, alleviate diarrhea, and regulate the fecal microflora of weaned piglets [94]. It was reported that mulberry leaf extracts’ additives significantly improved the feed conversion efficiency, the activity of glutathione peroxidase and catalase, and improved immunoglobulins. Moreover, the addition of mulberry leaf extracts increased the relative level of *Roseburia*, while decreasing the level of *Lactobacillus*. *Lactobacillus* is a type of probiotic that is important for the growth performance and the health of animals [95]. *Roseburia* is an abundant butyrate-producing bacteria that plays a great role in intestinal health [96]. The changes of the intestinal microflora induced by using mulberry leaf extracts did not have negative impacts on weaned piglets. Moreover, the use of mulberry leaf extracts can enhance the growth performance and antioxidant capacity, and alleviate diarrhea.

Fan et al. [97] also found that supplementary mulberry non- or fermented mulberry leaf powder could improve the growth performance and antioxidant activities, increase the fat deposition in muscle and reduce the pork rancidity and backfat thickness. Fermented mulberry leaves have better effects on improving the growth performance, meat quality, and antioxidant capacity than non-fermented mulberry leaves. Zhang et al. [98] evaluated the effect of fermented mulberry powder on the growth performance and meat quality of fattening pigs. The results indicated that, in comparison to the control group, the supplementation of fermented mulberry powder could increase the average daily gain, improve the pork quality by reducing the level of saturated fatty acids, and increase the content of unsaturated fatty acids. The addition of fermented mulberry leaves to the diet can effectively reduce the adverse effects of non-fermented mulberry powder on the intestines of pigs [99]. Fermented mulberry leaves added to the diet can improve antioxidant capacity by reducing total antioxidant capacity (T-AOC) in the serum. Ding et al. [100] also clarified that fermented mulberry leaves could enhance meat quality, reduce average backfat thickness and the content of total cholesterol in the serum. A dose of 9% had the best effect on growth performance of all the doses tested. In comparison to the control group, the ratio of feed to gain from 1 to 50 days was increased by 19%.Moreover, mulberry leaves also exhibited anti-obesity benefits by elevating the leptin-stimulated lipolysis, which provides a theoretical foundation for the development of mulberry leaves as anti-obesity drugs [101].

### 4.3. Ruminants

#### 4.3.1. Effects on Calf Health

Neonatal calf diarrhea (NCD), also known as calf scours or enteritis, is a gastrointestinal disease of pre-weaned calves that is associated with a high mortality [102]. Nutritional strategies play an important role in preventing the incidence of NCD in calves [103]. Table 5 summarizes the main effects of the mulberry tree extracts on the ruminant production.

Bi et al. [104] reported that feeding mulberry leaf flavonoids alone or combined with yeast (*Candida tropicalis*) improved the growth performance, reduced the fecal score, reduced the diarrhea rates, and regulated the intestinal microbiota in pre-weaned calves challenged with *E. coli* K99. In contrast to the calves challenged with *E. coli* K99, calves fed flavonoids or yeast had a higher relative abundance of *Prevotella*, *Lactobacillus*, and *Enterococcus*, and had lower numbers of *E. coli* K99 in jejunum digesta. 

Kong et al. [105] also showed that feeding mulberry leaf flavonoids along with yeast (*Candida tropicalis*) not only enhanced the growth performance and fecal score, but also improved the concentrations of immunoglobulin G (IgG) and immunoglobulin A (IgA) in the serum. Previous research has also reported a beneficial effect of mulberry leaf flavonoids on the fecal score and ADG [106]. The reduction of fecal scores happened simultaneously with the decrease of blood SOD and glutathione peroxidase (GSH-Px), in response to the mulberry leaf flavonoid supplementation. In addition, the lower mucosal thickness in the abomasum and duodenum was observed in calves fed mulberry leaf flavonoids, which might result in the greater absorption of nutrients.

#### 4.3.2. Effects on the Rumen Development and Rumen Microbiota

The rumen is a primary digestive organ of ruminants and contains diversified microorganisms [107]. The rumen and symbiotic bacteria play an important role in the growth performance and health status of ruminants [108]. Current studies show that early feeding regimes and nutrition strategies are associated with rumen development and the establishment of rumen microflora [109,110].

One study has reported the effect of mulberry leaf powder on the development of rumen epithelium in fattening Hu sheep [38]. The result revealed that the supplementary feeding of mulberry leaf powder can improve the development of rumen papillae and stratum basale in fattening Hu sheep. Additionally, it also reported that the supplementation of mulberry leaf powder improved the nutrient digestibility and microbial protein concentration, and tended to decrease the concentration of ammonia. Tan et al. [111] reported that the dietary supplementation with mulberry leaf pellets at 600 g/hd/d could improve the dry matter intake, ruminal NH_3_-N, and cellulolytic bacteria, thus improving the rumen ecology in beef cattle fed with rice straw. Wang et al. [112] also proved that mulberry leaves keep the rumen healthy, and have a potential positive impact on goat health by reducing the concentrations of leptin in the serum. In addition, supplementing the mulberry leaf flavonoids improved the activity of α-amylase in ruminal digesta and the activity of protease in abomasal digesta in calves [106].

A study has reported that mulberry leaf silage could increase the abundance of *Bifidobacterium*, *Lactobacillus*, and *Schwartzia*, and reduce the abundance of *Ruminococcaceae UCG-010,* and *Lachnospiraceae_XPB1014_group* [113]. These modified rumen microbial taxa were associated with the increased antioxidant and immunomodulatory capacity of lambs. Likewise, Li et al. [114] demonstrated that the supplementation of mulberry branch and leaves silage at 5–10% could improve the rumen microflora and fermentation, increase the abundance of fiber-digesting, propionic acid synthesis and milk fat-related microorganisms, thus improving the milk yield of dairy cows. Kong et al. [105] reported that the dietary addition of mulberry leaf flavonoids and yeast, either separately or together, was able to increase the concentration of the total volatile fatty acid and molar proportion of propionate during the pre-weaning and overall periods.

Methanogens exist in the rumen, and they can produce enteric CH_4_ through the hydrogenotrophic, acetoclastic, and/or methylotrophic pathways [115,116]. Enteric CH_4_ is considered to be a key contributor to climate change, which has aroused wide public concern [117]. Chen et al. [118] showed that the dietary supplementation of mulberry leaf flavonoids can improve the nutrients and energy utilization of sheep, and thus reduce CH_4_ emissions. Another in vivo study reported that mulberry leaf flavonoids improved the digestibility of organic matter, the content of volatile fatty acids in rumen, and the populations of *Fibrobacter succinogenes* [119]. The dietary supplementation of mulberry leaf flavonoids can also reduce the daily CH_4_ output by decreasing the ruminal populations of protozoans and methanogens.

Collectively, these findings indicate that mulberry extracts can potentially regulate the rumen development and rumen microbiota, subsequently improving the nutrient utilization and growth performance of ruminants while reducing the CH_4_ emissions.

#### 4.3.3. Effects on the Growth Performance and Immune Function

Mulberry leaves are rich in protein and can be used as a potential supplement of fermentable energy and protein for ruminants [26]. Liu et al. [120] reported that mulberry leaves may be used as a protein supplement to ammoniated straw diets, completely replacing rapeseed meals. Meanwhile, the mulberry leaves supplementation increased the feed intake and reduced feed costs. Mulberry leaves can partially replace lucerne hay, concentrates, and oilseed cakes [28,121], and have a positive interaction with oat hay on the fiber and crude protein digestibility of sheep [122]. For Vietnamese cattle, mulberry leaves efficiently serve as a supplemental source of protein to replace cottonseed, which is a more expensive animal feed ingredient. When using mulberry leaves as a supplement in the diet of Vietnamese cattle, it can replace cottonseed without any negative effect on the growth performance, but reduce the feed consumption to gain ratio by 8–14% [24]. Previous studies [123,124] showed that the supplementation of ensiled mulberry leaves (EML), sun-dried mulberry fruit pomace (SMFP), or corn grain and cottonseed meals in steer diets have similar effects on the performance, blood biochemical parameters, carcass characteristics, and rumen microbial composition. However, steers fed the SMFP diet had lower ammonia N in the rumen and lower intramuscular fat content [123]. Multiple studies found that mulberry silage was a cost-effective protein alternative for sheep [125,126] and lambs [127]. Sun et al. [128] found that partially replacing wildrye hay with mulberry leaves in the diet of sheep, improved the growth performance and carcass characteristics.

Zhang et al. [129] found that the mixture of mulberry leaf flavonoids and *Candida tropicalis,* as feed additives, can synergistically improve the growth performance and nutrient digestibility of calves. Mengistu et al. [16] indicated that the meal of dried mulberry and *vernonia* mixed leaves can substitute noug seed cake as a protein supplement, resulting in higher dry matter and nutrient intakes and body weight gain of yearling Bonga sheep.

Cheong et al. [17] reported that mulberry silage may be useful as a functional feed by improving the antioxidant activity of beef cattle. It was found that supplementation of mulberry silage increased the activities of glutathione, superoxide dismutase, glutathione peroxidase, catalase, and glutathione-S-transferase, compared to the control group. Moreover, the mulberry silage group exhibited a stronger radical scavenging activity, and the activity was dose-dependent. The addition of paper mulberry silage to the diets of dairy cows was able to increase the concentration of immune globulin, catalase, superoxide dismutase, and the total antioxidant capacity in the serum, thus improving the immune and antioxidant function of dairy cows [130,131]. Ouyang et al. [132] also found that the dietary mulberry leaf powder supplementation enhanced the antioxidant capacity of fattening Hu lambs by decreasing the alanine transaminase (ALT) and aspartate transaminase (AST) levels in serum, and stimulating the AOE gene expression.

#### 4.3.4. Effects on Animal Products

Mulberry tree products had a great potential for improving the milk production levels [133]. Venkatesh et al. [134] evaluated the changes in protein, carbohydrate, and fat content in the milk of cows and goats before and after the mulberry leaves feeding. Following feeding mulberry leaves for 60 days, the milk protein content of dairy cows and goats increased by 17.41% and 91.41%, respectively, while the carbohydrate content in milk of cows and goats decreased by 11.50% and 19.91%, respectively. Further, the lipid content in cow milk was increased by 36.36% and in goat milk by 39%, after feeding mulberry leaves for 60 days. Si et al. [130], in his experiment, fed 10% or 15% of paper mulberry silage to dairy cows, which resulted in a reduction of the milk somatic cell count and an increased content of polyunsaturated fatty acid in the milk. 

Another study has also noted that mulberry silage had a positive impact on the composition of fatty acids and amino acids in meat [135]. Sun et al. [128] concluded that replacing 24% of the conventional forage with mulberry leaves, in the diet of sheep, increased the intramuscular fat and pH of the 24 h *longissimus dorsi*. In addition, substituting the conventional forage with mulberry leaves decreased the content of saturated fatty acid and increased the content of monounsaturated fatty acid in the *longissimus dorsi* muscle of sheep. Another study found that the mulberry leaf powder supplementation could improve the color, tenderness, and water holding capacity of the *longissimus lumborum* muscle [132].

### 4.4. Fishes

Mulberry tree extracts can be used as a drug against fish parasites. The research progress of the application of the mulberry and its extract in fish production has been summarized in Table 6. The antiprotozoal activity of the mulberry root bark extracts was determined by Fu et al. [136] and their results illustrated that acetone and ethyl acetate extracts had a good killing effect on the non-encysted tomonts and theronts of *Ichthyophthirius multifiliis*. Liang, et al. [137] also found that two flavonoids from the mulberry root bark have the potential to control *Ichthyophthirius multifiliis*.

Mulberry tree extracts can enhance the antioxidant, anti-inflammatory, and immune activity of fish, thereby improving the resistance to diseases. Mulberry tree extracts have evident antioxidant and anti-inflammatory effects and have been added to many medicinal herb preparations [50], and thus have the great potential to enhance the immune activity of aquatic animals. In in vitro and in vivo studies, water extracts of the mulberry can attenuate the LPS-induced inflammatory and oxidant response in RAW 264.7 macrophages and zebrafish. The water extract of the mulberry attenuated the levels of pro-inflammatory mediators and the accumulation of ROS by suppressing the LPS-induced nuclear trans-localization of the NF-κB and MAPKs activation [138]. Yilmaz et al. [139] provided 2% black mulberry syrup to Nile tilapia, which could improve the growth performance, antioxidant status, immune activity, and disease resistance against *Aeromonas veronii*. Mulberry tree extracts enhanced the activities of antioxidant enzymes and the expression levels of the immune-related genes in the serum and spleen of fish. The result of an in vivo mechanistic study indicated that the mulberry leaves extract exhibited therapeutic effects against *Aeromonas hydrophila*, which was closely associated with its antioxidant and immune enhancing functions [140]. 

The economic benefit quality of aquatic products was highly related with their muscle quality. A recent study revealed that the supplementation of 10% paper mulberry can improve the muscle quality by increasing the muscle hardness, decreasing the fat accumulation and the muscle fiber diameter, at the expense of reducing the growth performance [141]. Moreover, the dietary supplementation of mulberry leaves decreased the cadmium residues in the liver of *Gobiocypris rarus* [142] and the finding may provide new strategies for mitigating the risks of heavy metals to human health.

## 5. Conclusions

Mulberry tree products contain a variety of bioactive substances with diverse biological functions, such as anti-oxidation, anti-inflammatory, anti-microbial, and anticancer properties. Therefore, it has the extensive potential for improving the health and productivity of food animals. In addition, mulberry tree products may be used as a supplementary protein source for animals. At present, there are six main application methods of the mulberry tree, including the fresh mulberry tree, mulberry tree extract, fermentation, or silage, mixed with traditional Chinese medicines, and mulberry powder. In animal production, the effects of the mulberry tree or its extracts on the growth performance and health may be related to its antioxidant, anti-inflammatory, and antimicrobial activities. At present, the mulberry tree used in livestock production is generally whole material rather than single compounds. The complexity and variability in the chemical composition of the mulberry tree make it arduous to clarify its mechanisms of action, and more experiments about the monomer compounds existing in the mulberry tree would be required. Most recent studies on the bioactivity and pharmacological effects of mulberry extracts are mostly carried out in laboratory animals, while few studies have been conducted on animal production. Therefore, it is essential to study further the promotion effects of the mulberry bioactive compounds on the animals’ performance, in order to provide a theoretical foundation and guidance for the application in animal husbandry production.

## Figures and Tables

**Figure 1 animals-12-03541-f001:**
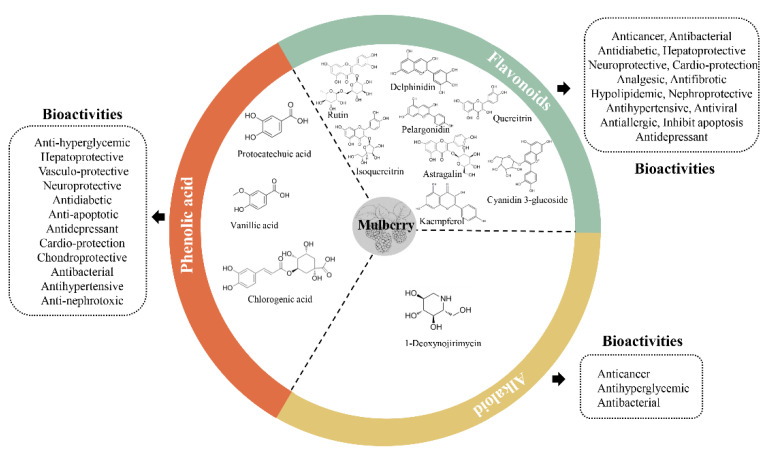
Main mulberry components and their bioactivities.

**Table 2 animals-12-03541-t002:** Trace element composition of the mulberry fruits (mg/100 g).

Authors	Species	Source	Country	Season	Fe	Zn	Ca	Mg,	K	Na	Mn	Cu
Ercisli et al., 2007	*Morus alba* L.	Fresh	Turkey	-	4.2	-	152	106	1668	-	-	-
	*Morus rubra*				4.2	-	132	106	922	-	-	-
	*Morus nigra*				4.2	-	132	115	834	-	-	-
Koca et al., 2008	*Morus latifolia*	Fresh	Turkey	-	2.85	0.52	88.9	19.4	210.75	11.89	0.35	0.31
Akbulut et al., 2009	*Morus nigra*	Freeze-dried	Turkey	-	3.72	0.96	304.4	95.9	1086.0	11.5	-	0.41
	*Morus rubra*				1.22	1.16	443.7	103.3	1526.9	21.8	-	0.65
	*Morus alba* L.				5.29	0.90	292.7	90.4	1310.2	21.5	-	0.68
Sheng et al., 2009	*-*	Fresh	China	-	1.17	0.14	38.89	12.21	234.65	16.1	0.03	0.04
Imran et al., 2010	*Morus alba* L.	Fresh	Pakistan	Spring	73.0	50.20	576	240	1731	280	-	-
	*Morus nigra*				77.6	59.20	470	240	1270	272	-	-
	*Morus laevigata* (Large white fruit)				48.6	53.40	440	360	1650	260	-	-
	*Morus laevigata* (Large black fruit)				63.6	50.80	576	240	1644	264	-	-
Altundag et al. 2011	*Morus alba* L.	Dry	Turkey	-	37.45	4.47	-	-	-	-	4.36	1.28
		Wet			40.80	4.38	-	-	-	-	4.25	1.31
		Microwave			36.17	3.78	-	-	-	-	4.16	1.23
Sánchez-Salcedo et al., 2015	*Morus alba*	Fresh	Italy	-	2.82–4.67	1.49–1.96	190–370	120–190	1620–2130	10	1.23–1.94	0.45–0.64
	*Morus nigra*				2.39–3.71	1.57–2.25	210–430	130–190	1480–2170	10	1.23–1.81	0.28–0.52
Jiang et al., 2015	*Morus alba* L.	Fresh	China	Summer	6.96	0.21	71	32.5	239	6.2	0.31	0.10
	*M. alba var. tatarica* L.				11.40	0.32	124	55.8	350	6.5	0.70	0.13
	*Morus nigra*				11.90	0.10	113	36.9	297	5.9	0.40	0.10
Sun et al., 2018	-	Fresh	China	Spring	7.72–30.13	4.06–10.58	180.61–423.30	13.96–33.38	87.70–208.44	-	-	0.04–0.50

Iron (Fe); Zinc (Zn); calcium (Ca); magnesium (Mg); kalium (K); sodium (Na), manganese (Mn); copper (Cu).

**Table 3 animals-12-03541-t003:** The application of mulberry tree extracts on poultry.

Authors	Species	Parts	Sources	Animal	Age of Animals	Dose or Concentration	Main Effects
Jang et al., 2008	-	Leaves	Mixed herbs containing mulberry leaf	Broiler	1–35 d	0.3 and 1.0%	Increased the antioxidative potential and overall preference of breast meat during cold storage
Islam et al., 2015	*Morus alba* L.	Leaves	Mulberry leaf powders	Broiler	1–42 d	2.5, 3.5, and 4.5%	Improved the performances and had a positive effect on the serum lipids
			Mulberry leaf powder extract			3.50%	
Lin et al., 2017	*Morus latifolia*	Leaves	Mulberry leaves	Laying hens	22–34 weeks	0.5, 1, and 2%	Modulated the antioxidative status of laying hens and improved their production performance and egg quality
Chen et al., 2019c	-	Leaves	Mulberry leaf powders	Broiler	120–157 d	4%	Changed the composition of the gut microbiota
Chen et al., 2019b	*Morus alba* L.	Leaves	Mulberry leaf extract	Egg Chickens	14–21 d	Orally administered 0.5, 1, 2, 4, 8, and 12 mg	Increased the serum Newcastle disease antibody titers, enhanced the immune effect of the vaccine, caused significant weight gain, and enhanced the intestinal and tracheal mucosal immune function
Ding et al., 2021	-	Leaves	Fermented mulberry leaf powder	Broilers	1–56 d	3, 6, and 9%	Improved the digestion and absorption of nutrients, growth performance, slaughter performance, and meat quality
Huang et al., 2021	-	Leaves	Mulberry leaf extract	Hens	60–68 weeks	0, 30, and 60 mg/kg	Improved the reproduction performance of aged breeder hens through improving the ovary function and hepatic lipid metabolism
Huang et al., 2022	-	Leaves	Mulberry leaf extract	Hens	60–68 weeks	0, 30, and 60 mg/kg	Ameliorated the eggshell quality of aged hens by improving the antioxidative capability and Ca deposition in the shell gland of the uterus

**Table 4 animals-12-03541-t004:** The application of mulberry tree extracts on swine.

Authors	Species	Leaves	Sources	Animal	Test Duration	Dose or Concentration	Main Effects
Li et al., 2012	-	Leaves	Mulberry leaf	Finishing pigs	65 d	10 and 20%	Does not affect the growth rate of finishing pigs, it can decrease the leaf lard percentage and back fat thickness, increase the contents of inosinic acid and fat in muscle
Song et al., 2016	-	Leaves	Mulberry leaf powder	Finishing pigs	50 d	5, 10, 15, and 20%	Has less effect on the growth rate of finishing pigs, while it can improve the meat quality
Zhang et al., 2016	-	-	Fermented feed mulberry powder	Fattening pigs	66 d	15%	Increased the average daily gain of pigs, increased the content of unsaturated fatty acids in pork, reduced the content of saturated fatty acids to a certain extent, improved the quality of pork, and realized the health care function of pork
Liu et al., 2019	*Morus latifolia*	Leaves	Mulberry leaves	Finishing pigs	50 d	3, 6, 9, and 12%	Mulberry in the diet at <12% is an effective feed crop to improve the meat quality and the chemical composition of the muscle without negatively affecting the growth performance
Chen et al., 2019a	-	Leaves	Mulberry leaves	Finishing pigs	67 d	3, 6, and 9%	May improve the muscle quality of pigs by modulating the expression of several key genes
Zeng et al., 2019	*Morus alba* L.	Leaves	Mulberry leaves	Finishing pigs	85 d	15%	Reduced the growth performance and carcass traits, but improved the meat quality of finishing pigs possibly through the change of the myofiber characteristics, the enhancement of the antioxidative capacity and the increase of intramuscular fat
Ding et al., 2019	Forage Mulberry	Forage mulberry	Fermented forage mulberry powder	Ningxiang Pigs	75 d	9, 12, and 15%	Improved the antioxidant performance, ameliorated the intestinal micro environment, and reduced the negative effects of mulberry powder on the intestine
Chen et al., 2020	*Broussonetia papyrifera*	Leaves	Paper mulberry leaf extract	Weaned Piglets	15 d	150 and 300 g/t	Increased the growth performance and the antioxidant capacity, reduced the occurrence of diarrhea, enhanced the immune functions and disease resistance, and affected the composition of fecal microflora
Fan et al., 2020a	*Morus nigra* L.	Leaves	Non- or fermented mulberry leaf powder	Finishing pigs	45 d	5%	Promoted the feed conversion ratio, reduced the backfat thickness, increased the fat deposition in the muscle, and reduced the rancidity of pork
Fan et al., 2020b	*Morus nigra* L.	Leaves	Mulberry leaves	Finishing pigs	45 d	5%	Have obvious anti-obesity effects
Liu et al., 2021	*-*	Leaves	Mulberry leaf powder	Finishing pigs	50 d	3, 6, 9, and 12%	Enhanced the serum antioxidant property, increased the polyunsaturated fatty acid content, and inhibited lipid oxidation by regulating gene expression levels of lipid metabolism and mitochondrial uncoupling protein in muscle tissue
Chen et al., 2021	*Morus alba* L.	Leaves	Mulberry leaves	Finishing pigs	31 d	4%	Promoted the growth performance, improved carcass traits, better quality of pork, and improved meat flavor

**Table 5 animals-12-03541-t005:** The application of mulberry tree extracts on ruminants.

Authors	Species	Parts	Sources	Animal	Test Duration	Dose or Concentration	Main Effects
Roothaert et al., 1999	*Morus alba*	Leaves and succulent twigs	Mulberry leaves and succulent twigs	Dairy heifers	14 d	Supplements at 25% of the estimated daily dry matter intake	Had a higher voluntary intake and a higher potential of milk production than the cassava tree (*Manihot glaziovii*), and fresh leucaena (*Leucaena diversifolia*)
Liu et al., 2001	*Morus alba*	Leaves	Mulberry leaves	Sheep	75 d	Replace 25, 50, 75, and 100% rapeseed meal	Mulberry leaves may be used as a proteinsupplement to ammoniated straw diets to fully substitute for rapeseed meal
Doran et al., 2007	*Morus alba* L.	Leaves	Mulberry leaves	Sheep	14 d	20 g DM/kg BW per day	Mulberry appears to be an excellent forage with many qualities, comparable and in some cases, superior to alfalfa. In addition, the combination of mulberry foliage with such a low-quality roughage as oat hay could produce a positive interaction on the protein and neutral detergent fiber digestibility
Kandylis et al., 2009	*Morus latifolia*	Leaves	Mulberry leaves	Sheep	19 d	Fed the prescribed ration	Mulberry leaves have an appreciable potential as a protein source in sheep feeding
Vu et al., 2011	*Morus alba* L.	Leaves	Mulberry leaves	Cattle	15 d	5, 10, and 15%	Improved digestibility of the crude protein and organic matter
Salinas-Chavira, et al., 2011	*Morus alba* L.	Leaves	Mulberry leaves	Lambs	10 d	2.5% or 5%	Partly replace the expensive protein-rich ingredients, such as oilseed cakes and help increase the profitability of lamb fattening
Tan et al., 2012	-	Leaves and stems	Mulberry leaf meal	Beef cattle	84 d	0, 200, 400 and 600 g/hd/d	Improved the dry matter intake, ruminal NH_3_-N, and cellulolytic bacteria thus improved the rumen ecology in beef cattle fed with rice straw
Jeon et al., 2012	*Morus alba* L.	Leaves and stems	Mulberry silage	Hanwoo (Bos taurus coreanae) steer	36 d	10%	Improved the hematological traits and composition of fatty acids and amino acids
Zhou et al., 2014	-	Leaves	Ensiled mulberry leaves	Finishing steer	112 d	8%	Had similar effects to corn grain and cottonseed meals on the steer performance, blood biochemical parameters, and carcass characteristics, with the exception of ruminal VFA concentrations and lower intramuscular fat content
		Fruit	Sun-dried mulberry fruit			6.30%	
Yulistiani et al., 2015	*Morus alba* L.	Leaves	Mulberry leaves	Sheep	22 d	50% or 100% mulberry replaced with rice bran and urea	Had a similar effect to urea rice bran supplementation on the dry matter intake, nutrient digestibility and N utilization that create an efficient rumen ecosystem and microbial protein supply
David et al., 2015	-	Mulberry branches	Mulberry silage	Sheep	150 d	100% sugarcane silage, 75% sugarcane silage plus 25% mulberry branches silage, and 50% sugarcane with 50% mulberry branches silage	The protein in mulberry silage replaced the protein offered in the concentrate, making this mulberry silage a cost-effective alternative
Venkatesh et al., 2015	-	Leaves	Mulberry leaves	Goat and cow	60 d	Fed a basal diet supplemented daily with mulberry leaf flavonoids (5.0%, *w*/*w*; 3 g/calf) and then orally challenged with *E. coli* K99	Enhanced the quality and quantity of goat and cow’s milk
Niu et al., 2016		Leaves	Ensiled mulberry leaves	Finishing steers	-	8%	The partial replacement of corn grain and cottonseed meal with ensiled mulberry leaves or sundried mulberry fruit pomace had no substantial effect on the ruminal microflora composition
		Fruit	Sundried mulberry fruit pomace			6.30%	
Bi et al., 2017	-	Leaves	Mulberry leaf extract	Pre-weaned calves	36 d	Fed a basal diet supplemented daily with mulberry leaf flavonoids (5.0%, *w*/*w*; 3 g/calf) and then orally challenged with E. coli K99 (30 mL; 1.0 × 109 CFU/mL)	Had a higher average daily weight gain and feed efficiency, reduced days of diarrhea, improved intestinal health, and beneficially manipulated the intestinal microbiota in pre-weaned calves
Ma et al., 2017	-	Leaves	Mulberry leaf extract	Sheep	14 d	2 g/head/day	Improved the digestibility of the organic matter and reduced the enteric methane (CH4) output by inhibiting the populations ofmicrobes involved in methanogenesis
					42 d		
Alpízar-Naranjo et al., 2017	*Morus alba* L.	-	Mulberry foliage	Lambs	84 d	Inclusion of MLP at 0.75 and 1%in a concentrate diet	Replaced the imported concentrate while increasing the level of substitution of mulberry in the ration of fattening lambs
Zhang et al., 2017	-	Leaves	Mulberry leaf extract	Calves	60 d	Starter at 2 g/d per calf before weaning, or 4 g/d per calf after weaning	Increased growth performance and nutrient digestibility
Wang et al., 2018	*Morus alba* L.	Leaves	Mixed extract of mulberry leaf	Preweaning calves	35 d	3 g/calf per day	Improve the antioxidant function and reduce the incidence of oxidative stress after challenge with *E. coli* in 28-d-old preweaning calves
Ouyang et al., 2019	-	Leaves	Mulberry leaf powder	Fattening Hu sheep	84 d	0, 15, 30, 45, and 60%	Promoted the nutrient digestibility in the rumen and promoted the development and the metabolic properties of the rumen epithelium
Kong et al. 2019	*Morus alba* L.	Leaves and stems	Mulberry leaf extract	Calves	60 d	3 g/d	Increased the proportion of propionate among rumen fermentation products, increased the growth performance and decreased the fecal scores
Mengistu et al., 2020	*Morus indica*	Leaves	Mixed meal containing mulberry leaf	Sheep	90 d	The dietary treatments were a replacement of the protein in noug seed cake of concentrate mix at 25, 50, 75, and 100% proportions with dried mulberry and vernonia mixed leaves’ meal	Resulted in the optimum performances in terms of feed intakes, digestibility of the feeds, and the growth performances
Wang et al., 2020	-	Leaves	Ensiled mulberry leaves	Goats	70 d	10, 15, and 20%	Keeps rumen healthy (changes in the ruminal microbiota and fermentation parameters) and results in a potential positive impact on the host health
			Sun-dried mulberry leaves			10 and 15%	
Sun et al., 2020	-	Leaves	Mulberry leaves	Sheep	65 d	8, 24 and 32% replace Chinese wildrye	Had a beneficial influence on the growth performance, blood metabolites, and carcass characteristics
Hao et al., 2020	*Broussonetia papyrifera*	Leaves and stems	Paper mulberry silage	Holstein dairy cows	28 d	4.5 and 9.0%	Enhanced the antioxidant capacity and immunity of dairy cows, but did not influence the milk yield, dry matter digestibility, and fecal bacteria composition
Wang et al., 2021	*Morus alba* L.	Leaves	Mulberry leaf silage	Lambs	84 d	20% mulberry leaf silage	Improved the antioxidant capacity and immune function, and modified the rumen microbial community
Li et al., 2022	-	Branch and leaves	Mulberry branch and leaves silage	Cows	56 d	5 and 10%	Modulated the rumen microbiota and fermentation, increased the abundance of fiber-digesting, propionic acid synthesis, and milk fat-related microorganisms, thus improving the milk yield in dairy cows

**Table 6 animals-12-03541-t006:** The application of mulberry tree extracts on fishes.

Authors	Species	Parts	Sources	Animal	Test Duration	Dose or Concentration	Main Effects
Fu et al., 2014	*Morus alba* L.	Root bark	Mulberry root bark extract	Grass carp	96 h	2,4, and 8 mg/L	Protected fish from *I. multifiliis* infection
Liang et al., 2015	*Morus alba* L.	Root bark	Mulberry root bark extract	Grass carp	96 h	0.125, 0.25, 0.5, 1, and 2 mg/L	Controls *Ichthyophthirius multifiliis*
Kwon et al., 2017	*Morus alba* L.	Leaves	Mulberry leaves	Zebrafish larvae	-	800 μg/mL	Exerted potent anti-inflammatory and antioxidant effects in zebrafish
Yilmaz et al., 2020	*Morus latifolia*	-	Mulberry extract	Nile tilapia (Oreochromis niloticus)	60 d	0.75, 1.5, 2.0, and 3.0%	Improved the growth performance, innate immune parameters, antioxidant related gene expression responses, and disease resistance against *Aeromonas veronii*
Xiong et al., 2020	-	Leaves	Mulberry leaves	Rare minnow (Gobiocypris rarus)	28 d	10 and 30%	Decreased the Cd residues in the liver
Neamat-Allah et al., 2021	*Morus alba* L.	Leaves	Mulberry leaves	Oreochromis niloticus	30 d	1, 3 and 5 g/kg	Protected tilapias from hemato-biochemical alterations and enhanced its immune feedback, antioxidant defense, and resistance against A. hydrophila
Tang et al., 2021	*Broussonetia papyrifera*	-	Mulberry stems and leaves	Grass Carp (Ctenopharyngodon idella)	56 d	5, 10, 15, and 20%	Improved the muscle quality through improving the muscle hardness, reducing the fat accumulation, and the muscle fiber diameter, at the cost of reducing growth performance

## Data Availability

Not applicable.

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
