# Peer review of "Protective Application of *Morus* and Its Extracts in Animal Production"

_animals, 2022, doi:10.3390/ani12243541_

Round 1
Reviewer 1 Report
The use of alternative feeds and feed additives (including mulberry) represents a wide practical and theoretical potential for the development of agriculture, primarily animal husbandry and aquaculture. Since they allow you to simulate the immune status and obtain high-quality livestock products and ensure productive longevity. In my humble opinion, the article deserves acceptance into the journal for further publication.
However, in the process of consideration, some questions arose recommendations for authors to improve the content of the manuscript.
Line 80-82: It is necessary to edit the sentences. A misconception is created (scorbic acid (200-280 mg/100 g), β-car-81 otene (10,000-14,688 µg/100g), do not belong to mineral substances). Similarly, in Table 2, carotene and ascorbic acid do not belong to mineral elements.
The authors do not need to decipher abbreviations every time (for example, line 77: dry matter (DM; 19.8-30.40%); line 84 and so on). One mention before the abbreviation is enough.
Line 79: What energy your use? Gross or metabolizable?
In Table 1, it is recommended to verify which indicators you provide (which fat is crude fat or essential extract, which fiber is crude)?
There are a lot of rows in the tables that are not filled in, it is recommended to review the tables throughout the test. Make them more informative. For example, separately for minerals, vitamins, anti-nutrients, etc.
For example, in Table 2, tannic acid has data everywhere, according to Ascorbic acid, β-Carotene indicators only one author has.
In the table 3 Parts can be removed, because it is clear from the title of the chapter.
It is enough to specify the units of measurement in the header of the table (if they match everywhere in the table and all elements (Table 4)).
Please select one style in the tables (abbreviated or completely) in Table 4, etc.
In section 4.1. Poultry, it is better to describe the results of the study in more detail.
Author Response
Dear reviewer:
Thank you for your comments concerning our manuscript entitled “Protective Application of Morus and Its Extracts in Animal Production”. Those comments are all valuable and very helpful for revising and improving our manuscript. We have studied comments carefully and have made corrections in the manuscript which we hope meet with approval. In this revised version, changes to our manuscript within the manuscript were all highlighted as yellow and newly added references were highlighted by using red colored text. Point-by-point responses to the reviewer are listed below this letter. The main corrections in the paper and the responds to the reviewer’s comments are as following:
Line 80-82: It is necessary to edit the sentences. A misconception is created (scorbic acid (200-280 mg/100 g), β-car-81 otene (10,000-14,688 µg/100g), do not belong to mineral substances). Similarly, in Table 2, carotene and ascorbic acid do not belong to mineral elements.
Response: Thanks for your suggestion. In accordance with your question about filling the Table, we have deleted the Table 2.
The authors do not need to decipher abbreviations every time (for example, line 77: dry matter (DM; 19.8-30.40%); line 84 and so on). One mention before the abbreviation is enough.
Response: We have carefully corrected these errors throughout the manuscript.
Line 79: What energy your use? Gross or metabolizable?
Response: Thanks for your suggestion. We have revised “energy” to “gross energy” (Line 82).
In Table 1, it is recommended to verify which indicators you provide (which fat is crude fat or essential extract, which fiber is crude)?
Response: We have added the footnote in Table 1 as suggested (Lines 96-98). In table 1, there are 13 studies [22-25,27-30,32-34,37-38] measured fat. The studies of Srivastava et al., 2006 [22], Adeduntan et al., 2009 [27], Iqbal et al., 2012 [33], Flaczyk et al., 2013 [34] and Kang et al., 2020 [37] were described as crude fat. The rest of studies were described as ether extract. In addition, we have revised “fiber” to “crude fiber (CF)”, because the fiber measured in the studies was all crude fiber.
There are a lot of rows in the tables that are not filled in, it is recommended to review the tables throughout the test. Make them more informative. For example, separately for minerals, vitamins, anti-nutrients, etc. For example, in Table 2, tannic acid has data everywhere, according to Ascorbic acid, β-Carotene indicators only one author has.
Response: Thanks for your suggestion. When we collected data, we found that only one article measured the vitamins content of mulberry leaves. As for the anti-nutrient content of mulberry leaves, only tannic acid content was detected in the articles. No sufficient data were provided in the reviewed articles. We thus presented this part of data in text form and deleted the Table 2.
In the table 3 Parts can be removed, because it is clear from the title of the chapter.
Response: We have removed Table 3 as suggested.
It is enough to specify the units of measurement in the header of the table (if they match everywhere in the table and all elements (Table 4)).
Response: We have revised as suggested (Line 117).
Please select one style in the tables (abbreviated or completely) in Table 4, etc.
Response: Thanks for your suggestion. We have used abbreviations to represent the tables uniformly.
In section 4.1. Poultry, it is better to describe the results of the study in more detail.
Response: Thanks for your suggestion. We have described the results in more detail as suggested (Lines 184-187 and Lines 194-198).
Reviewer 2 Report
This review summarizes the nutrients, active compounds, and biological functions of mulberry, as well as the application in livestock production with an aim to provide a reference for the utilization of mulberry and its extracts in animal production. Strengths: ruminants are presented in detail - calf health, rumen and microbiome, growth indicators and immune function, impact on animal products. It's interesting for science.
The topic of the review is presented in detail, the topic of the review is relevant for raising animals without antibiotics, a gap in knowledge has not been identified, references are appropriate.
The scientific content of the manuscript is competently described and quite specific.
The review is clear, comprehensive and relevant to the field.
A similar review has not been published before, it is relevant and of interest to the scientific community.
The cited references are mainly recent (over the past 5 years) and current publications. An excessive number of self-citations was not noticed.
The statements and conclusions made are consistent and are confirmed by the listed quotations.
Tables and images are suitable and display the data correctly. They easy to interpret and understand.
Author Response
Dear reviewer:
Thank you for your comments concerning our manuscript entitled “Protective Application of Morus and Its Extracts in Animal Production”.
Reviewer 3 Report
The manuscript is well written and relevant to the animal production industry. I would like for the authors to specify what part or component of the mulberry tree they are referring to. I feel that just saying mulberry is not sufficient. The authors should also avoid repeating the details that are in tables in the text

Author Response
Dear reviewer:
Thank you for your comments concerning our manuscript entitled “Protective Application of Morus and Its Extracts in Animal Production”. Those comments are all valuable and very helpful for revising and improving our manuscript. We have studied comments carefully and have made corrections in the manuscript which we hope meet with approval. In this revised version, changes to our manuscript within the manuscript were all highlighted as yellow and newly added references were highlighted by using red colored text. Point-by-point responses are listed below this letter. The main corrections in the paper and the responds to the reviewer comments are as following:
The manuscript is well written and relevant to the animal production industry. I would like for the authors to specify what part or component of the mulberry tree they are referring to. I feel that just saying mulberry is not sufficient.
Response: Thanks for your suggestion. We have specified what part or component of the mulberry tree throughout the manuscript as suggested.
Line 76-90 and 100-109: Avoid repeating verbatim what is in the tables. Rather analyze or discuss the results in the table. Give context with respect to country, season etc
Response: Thanks for your suggestion. We have supplemented the corresponding messages in the table 1 and table 2, and added the corresponding text in Lines 113-116 as suggested.
Line 254: Is that a good or bad thing? Lactobacillus are generally considered good bacteria arent they?
Response: It is difficult to judge whether it is good or bad. The diverse microbial communities in gut comprise a complicated ecosystem in which microbe-microbial and microbe-host interactions influence host health (Proal, A. D. et al., 2017; Chen L. et al., 2020). Lactobacillus is a type of probiotic that is important for the growth performance and health of animals (Zhu, C. et al., 2022). Although the addition of mulberry leaves extracts decreased the level of Lactobacillus. Meanwhile, the relative level of Roseburia was increased, which is an abundant butyrate-producing Firmicutes plays a great role in intestinal health (Nie, K. et al., 2021). In addition, the reduction of Lactobacillus did not adversely affect weaning piglets. We have added the explanation in this manuscript (Lines 264-268).
References:
Proal, A.D. et al. “Microbe-microbe and host-microbe interactions drive microbiome dysbiosis and inflammatory processes.” Discovery medicine vol. 23,124 (2017): 51-60.
Chen L. et al. “Gut microbial co-abundance networks show specificity in inflammatory bowel disease and obesity.” Nature Communications. 11,1 (2020): 4018.
Zhu, C. et al. “A meta-analysis of Lactobacillus-based probiotics for growth performance and intestinal morphology in piglets.” Frontiers in Veterinary Science 2022, 9, doi:10.3389/fvets.2022.1045965.
Nie, K. et al. “Roseburia intestinalis: A Beneficial Gut Organism From the Discoveries in Genus and Species.” Frontiers in Cellular and Infection Microbiology 2021, 11, doi:10.3389/fcimb.2021.757718.
Line 292: ?? Not clear
Response: This sentence has been revised (Line 305).